# Quenched and Tempered Steels Welded Structures: Modified Gas Metal Arc Welding-Pulse vs. Shielded Metal Arc Welding

**Houman Alipooramirabad** [1,*]**, Neville Cornish** [2]**, Rahim Kurji** [1,2]**, Anthony Roccisano** [3,4] **and Reza Ghomashchi** [1]

1 School of Mechanical Engineering, The University of Adelaide, Adelaide, SA 5005, Australia
2 Australian Welding Solutions, Adelaide, SA 5039, Australia
3 Future Industries Institute, The University of South Australia, Mawson Lakes, SA 5095, Australia
4 The Australian Research Council (ARC) Industrial Transformation Training Centre in Surface Engineering for Advanced Materials (SEAM), Hawthorn, SA 5062, Australia
* Correspondence: houman.amirabad@gmail.com

**Abstract:** Quench and tempered (Q&T) steels are widely used for a diverse range of applications, particularly in the mining and defence industry, where wear and unconventional loading are common. Furthermore, they are particularly prone to hydrogen assisted cold cracking (HACC), imposing a more careful selection of consumables and requiring a comparably higher welder skill level to fabricate defect-free structures. Therefore, the cost of fabrication of welded structures is higher when the more preferred welding technique of shielded metal arc welding, SMAW, is employed. The introduction of the modified pulsed arc mode of depositions, a variation to pulsed arc deposition, has improved the productivity rates and can be utilised by welders with a greater skill variations. In this study, full-strength butt welds of Q&T steel (AS/NZS 3597 Grade 700), with the thickness of 20 mm, are fabricated under a high level of restraint using both conventional SMAW and modified pulse gas metal welding (GMAW-P). The study investigated the economic feasibility of the two deposition modes and the propensity to cracking for the welded joints under high restraint conditions. Utilising the modified GMAW-P resulted in 63% and 88% reduction in the 'Arc-On' time and the total normalised fabrication time, respectively. However, strict controls must be implemented, due to the increased propensity to lack of fusion-type defects, to optimise the welding procedure and mediate for such defects if GMAW-P is to provide a techno-economically beneficial alternative to conventional SMAW when welding Q&T steels.

**Keywords:** quench and tempered steel; shielded metal arc welding; modified pulse gas metal arc welding; Modified Welding Institute of Canada (MWIC) weldability test; hydrogen assisted cold cracking





## 1. Introduction

Quenched and tempered (Q&T) steels, which may be classified as high strength low alloy steels with respect to carbon and alloying elements concentration, have high strength and are resistant to abrasion. They are used in a range of applications from armoured structures in defence to pressure vessels and wear resistance earth moving machinery in mining. The selected Q&T steels are expected to have high toughness, wear resistance, tensile strength, notch toughness, ductility, and fatigue characteristics. In addition, they are expected to have good weldability, as welding is the main fabrication route for defence, mining and pressure vessels [1–3]. Although much research is devoted to the development and testing of armour grade materials [4–6], high strength armour steels are still widely adopted for the fabrication of safety-critical structures in the defence industry [7,8]. The application of Q&T steels in defence as armoured structures is due to their ability to withstand non-conventional loading conditions from threats such as ballistic projectiles and explosions. In Australia, the more popular Q&T steels are classified as bisalloys obtained from their manufacturer, Bisalloy® (https://www.bisalloy.com.au/, accessed on 16 January

2023) steel (Wollongong, NSW, Australia). Bisalloys are recommended for applications where wear is the main concern [9]. The superior weldability of these steels coupled with their high strength and wear resistance made them the prime choice in the mining and resources sector in Australia for the fabrication of equipment such as demolition and ground engaging tools, dump truck bodies, storage bins, hoppers, earthmoving buckets, drag line buckets, and wear plates Chutes where lighter weight, superior wear and impact resistance, combined with straightforward fabrication and weldability, are the prime factors for their selection. The selection of Q&T steels for pressure vessels are recommended in both European, EN 10028-6, and American, ASME, standards [10].

The welding of Q&T steels as the most preferred fabrication route is expected to alter the material microstructure as the microconstituent phases transform due to the heat generated during welding [3,11]. The resulted heat affected zone (HAZ) exhibits a softening effect [4,12] with consequences in compromising creep and fatigue properties and resulting in poor ballistic performance [13,14]. In addition to microstructural changes during welding, the issue of hydrogen dissolution in the weld metal (WM) and its diffusion during cooling (resulted from austenite to ferrite transformation) makes Q&T steels susceptible to hydrogen embrittlement specifically termed hydrogen-assisted cold cracking, HACC.

HACC also referred to as hydrogen induced cracking (HIC), delayed cracking or cold cracking, is a welding defect which in high strength low alloyed (HSLA) steels and Q&T steels may occur in the heat affected zone (HAZ) or in the weld metal (WM) of a welded joint [15,16]. Traditionally, hydrogen cracks were known to initiate in the coarse-grained HAZ region of the weld which were frequently described as root cracking, toe cracking, or under-bead cracking, depending on the crack locations [15]. These cracks may form immediately after welding or at some time after welding. The time to cracking after the welding is completed may range from a few minutes to several weeks. The greatest risk of HACC occurs after the weld has cooled to temperatures below approximately 200 °C [16]. Above this temperature, cracking is unlikely to initiate in HSLA or Q&T steels [15,17]. HACC is by far the most serious and dangerous imperfection in welded structures due to its least chance of detection and estimation of the time of activation [18]. The manifestation of HACC defect in welded structures is attributed to the accumulation of a critical concentration of hydrogen trapped in a susceptible microstructure that is subjected to a high magnitude of the tensile residual stresses [15,17,19,20]. However, the timing of the crack initiation is dependent on the incubation period of the residual hydrogen, i.e., reaching the critical concentration, which vary with the microstructure and the level of applied stress. The structural integrity of both the weld joint and the asset is threatened by the presence of the hydrogen cracks. This threat is even worsened by the applied loading conditions which these assets may encounter during their service life in severe working environments of the defence or mining industry. Therefore, considerable effort is required, particularly with the complex nature of delayed hydrogen cracking, to be employed during the design of welding procedures and fabrication to minimise the susceptibility of weldment to HACC. Shielded metal arc welding (SMAW) is the preferred welding technique used in the construction of major infrastructure structures including mining equipment, pressure vessels, and combat platforms and structures fabricated from Q&T steels [3,11,21–24]. To minimise the risk of hydrogen cracking, low hydrogen consumables are employed in the design of a welding procedure specifications (WPS) of the Q&T steel welds with optimised mechanical properties, such as toughness and strength [25,26]. However, this process has two major limitations including low deposition rate (inherently slow welding process) and the need for the highly skilled labour for joint fabrications with the SMAW process [27,28]. These two factors limit productivity and increase the overall cost associated with fabrication [29–31].

To address the productivity-associated limitations, there is a need to investigate the feasibility of an alternative welding process. In the context of the fabrication of safety-critical structures in the defence, mining, or energy transportation industry-pressure vessels, GMAW-P may be regarded as a cheaper alternative. It is important to highlight that

conventional gas metal arc welding (GMAW) is discounted as a result of the inability to control the characteristics of weld deposition independent of heat input [25,32–35]. Pulsed gas metal arc welding, however, is considered to be an exception. Moreover, the introduction of high clock frequency controllers into welding devices have facilitated the precise control of essential pulsed arc parameters such as the current rise and drop rates, the level and duration of the background and pulsed currents as well as the pulse frequency [33,36]. The controller's response allows for the adoption of different modulation types (I/I and U/I modulations) and regulation strategies to be utilised. This brings about low-spatter droplet transfer, high process stability, and the ability to deposit weldments with strict control of the heat input [32,33].

Synergic process variants such as GMAW-P, which involves a modified I-I-I-controlled, non-short-circuiting pulse integrate the characteristics of the classic pulse arc with those of the classic spray arc and are particularly beneficial for high productivity welding [33]. Nonetheless, there is limited published work in comparing the productivity gains of GMAW-P with SMAW. Furthermore, acknowledging the inherent risk of fusion-type defects (root and inter-pass fusion, lack of side wall) common with GMAW, there is a need to compare not only productivity but also the structural integrity of the fabricated joints deposited using similar consumables and under a range of comparable conditions.

In the present study, the comparative differences in productivity, microstructural and mechanical properties, and susceptibility to HACC of the weldments deposited using GMAW-P and conventional SMAW are investigated.

## 2. Experimental Procedures

The testing was performed on a 20 mm thick sections of AS/NZS 3597 Grade 700 (EN 10137-2 Grade S690Q) quench and tempered steel, Bisalloy 80 (https://www.bisalloy.com.au/, accessed on 16 January 2023). The Bisalloy 80 steel plates were provided by Bisalloy Steel Group Limited (Wollongong, NSW, Australia). The experiments were conducted over a range of welding parameters (i.e., typical heat input range) that would be expected during industrial fabrication of safety-critical structures as specified below.

### 2.1. Welding

To examine the technical integrity and economical advantages, the techno-economic feasibility, of employing GMAW-P over conventional SMAW when welding thick sections of Q&T steel, a three-stage experimental program was adopted.

#### 2.1.1. Stage 1: Productivity Testing

To examine the difference in the arc time and total fabrication time between the two deposition modes, multi-pass full strength single-sided butt welds were deposited on coupons with the length and width of 150 mm and 50 mm, respectively. Three test welds were deposited per target heat input, and an average of the 'arc on' time (AOT) and total time (TT) for deposition and were recorded and compared.

#### 2.1.2. Stage 2: Weldability Testing—HACC Susceptibility

To establish the susceptibility of Q&T Bisalloy 80 to HACC, the *Modified Welding Institute of Canada (MWIC) weldability test* [37] coupons were prepared, Figure 1, and multi-pass welds were deposited on the MWIC weldability test joint, Figure 2, using the same parameter range that was tested in Stage 1.

The MWIC test has been benchmarked as a quality procedure to ensure sound and crack-free in-field girth welding of gas transmission pipeline steel. A weldability test, representative of pipeline girth welding for small diameter thin-walled pipes, typically found in the Australian context, that subject a girth weld deposited into a V-prep groove to high restraint conditions similar to the in-field pipeline girth welding. The restraint in this weldability test is imposed through anchor welding of the parent plate to the base plate in all sides of the parent plate, with the restraint length of 25 mm as specified in Figure 1.

Furthermore, the run on/off tabs are designed to minimise the effects of the weld starting and end effects.

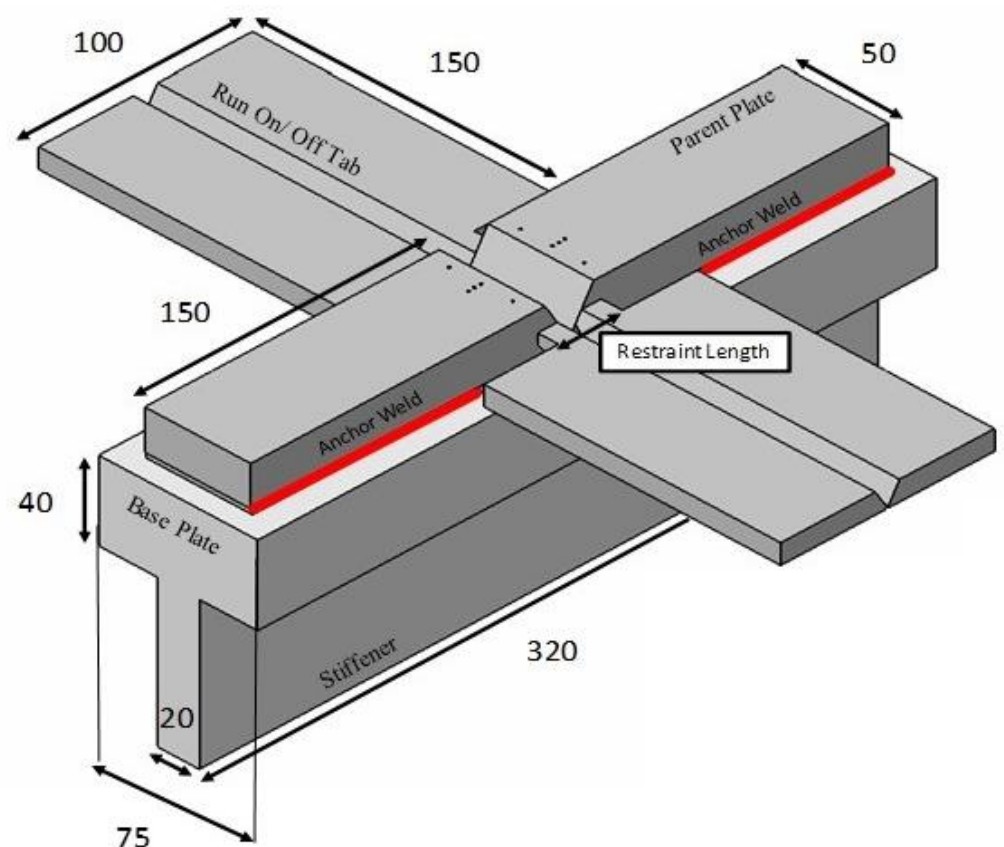

**Figure 1.** Geometry of the MWIC weldability test (all dimensions in mm). Reprinted/adapted with permission from [37], [Modified WIC test: An efficient and effective tool for evaluating pipeline girth weldability]; published by [Taylor & Francis Group], 2017.

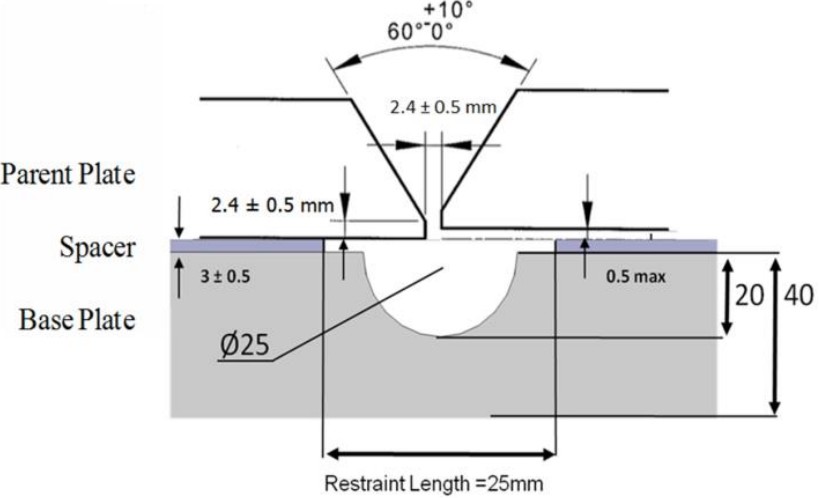

**Figure 2.** Dimensions of the single V-groove butt weld prep used for the MWIC weldability testing (all dimensions in mm). Reprinted/adapted with permission from [37], [Modified WIC test: An efficient and effective tool for evaluating pipeline girth weldability]; published by [Taylor & Francis Group], 2017.

The welded joint was removed from the MWIC specimens 24 h after the weld completion by milling the test assembly just inside the restraint length. The anchor welds were sawed off using a Struers water-cooled precision metallographic saw. The weld zone was assessed for cracking by examining six weld metal transverse cross-sections prepared for metallurgical inspection, Figure 3. A sample was defined as cracked, Figure 4, if a planner defect was visually identified on a sample surface when magnified at ×400, and the vertical length of the defect was greater than 5% of the height of the weld bead ($t_w$). The weld metal microstructure was examined using optical microscopy (Zeiss Axio Imager, Wetzlar, Germany) on metallographically prepared weldment cross-sections, (polished down to 1 µm diamond paste and in 2% nital solution etchant).

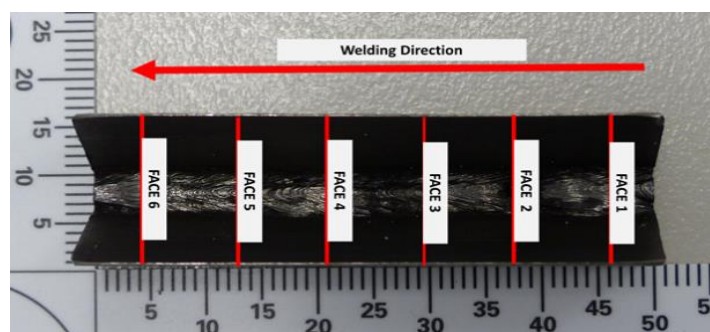

**Figure 3.** Weldability test piece with the location of the 6 faces for the examination under the optical microscopy at a magnification of ×400 highlighted. Reprinted/adapted with permission from [37], [Modified WIC test: An efficient and effective tool for evaluating pipeline girth weldability]; published by [Taylor & Francis Group], 2017.

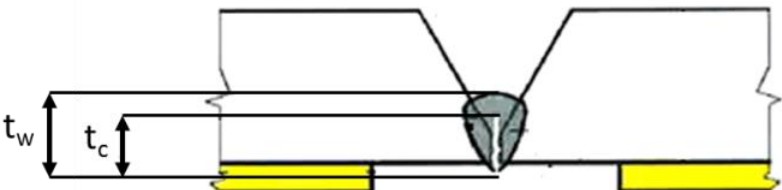

**Figure 4.** Schematic of a face of a test section. A sample is considered to be as cracked if the vertical length ($t_c$) of a linear defect is greater than 5% of the bead height ($t_w$). Reprinted/adapted with permission from [37], [Modified WIC test: An efficient and effective tool for evaluating pipeline girth weldability]; published by [Taylor & Francis Group], 2017.

2.1.3. Stage 3: Procedure Qualification—Mechanical Testing

To compare the mechanical properties of welds deposited using the conventional SMAW and novel GMAW-P, a single coupon 400 mm in length and 300 mm in width (prepared from two plates 400 mm long, 150 mm wide, and 20 mm thick welded using each process) was examined under a range of mechanical tests to qualify the procedures employed on the coupons to AS/NZS 1554.4:2014 (SP). Mechanical tests including hardness traverses, impact (Charpy V notch) tests at −29 °C, transverse tensile tests, transverse bend tests, and macroscopic examination were carried out in accordance with AS/NZS 2205:2003.

2.1.4. Welding Parameter Selection and Control

Weldability testing was carried out on a uni-directional, mechanised cradle, Lorch Track RL Pro, Lorch South Pacific, NSW, Australia (https://www.lorch.eu/, accessed on 28 November 2022) with the test specimen in a fixed horizontal position. For GMAW-P, the welding torch was secured in the cradle, and the trigger was manually operated to initiate and extinguish the welding arc, Figure 5. For SMAW, the electrode holder was manually manipulated. The arm of the mechanised cradle was used as a guide to govern travel speed.

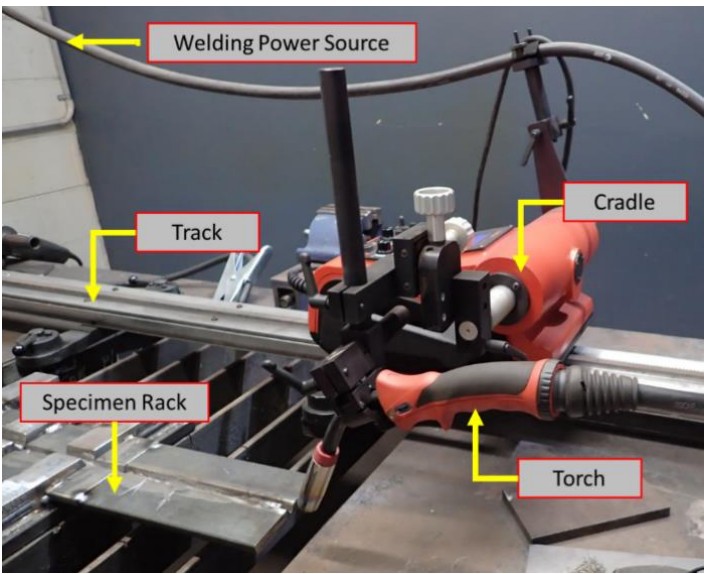

**Figure 5.** Experimental set up of MWIC weldability test. The specimen racks were adopted to hold 400 mm length sections for procedure qualification testing, 150 mm length coupons for productivity testing, and MWIC weldability tests for HACC testing.

The welding parameters employed during testing are summarised in Tables 1 and 2. The Lorch S3 Speed XT was used as the welding power source (https://www.lorch.eu/, accessed on 28 November 2022).

**Table 1.** Welding parameters—GMAW-P.

| Welding Specifications | | Welding Parameters | |
|---|---|---|---|
| Diameter of Wire | 1.2 mm | Current | 135–225 A |
| Direction | Flat (1G) | Travel Speed | 170–400 mm/min |
| Polarity | DC+ | Voltage | 21–30 V |
| Specification | ER 110S-G | Pre Heat Range | 25 °C |
| AWS Class | A5.28 | Heat Input Range | 0.48–2.19 kJ/mm |
| Shielding Gas | Ar 18% $CO_2$ (15–20 L/min) | Deposition Mode | GMAW-P (ISO 857 (Process No. 13)) |

**Table 2.** Welding parameters—SMAW.

| Welding Specifications | | Welding Parameters | |
|---|---|---|---|
| Direction | Flat (1G) | Current | 130–180 A |
| Size of Electrode | 3.2 & 4.0 mm | Voltage | 21–25 V |
| AWS Class | A5.5 | Travel Speed | 80–290 mm/min |
| Specification | E11018M-H4 (moisture resistant low-hydrogen coated electrode) | Heat Input Range | 0.41–2.22 kJ/mm |
| Polarity | DC+ | Pre Heat Range | 25 °C |

### 2.2. Materials Specifications

AS/NZS 3597 Grade 700 (EN 10137-2 Grade S690Q) steel plates was used for the fabrication of the MWIC and procedure qualification test coupons. Parent metal (PM) plates were sectioned from a single steel plate using the Techni INTEC-G2 1612 water jet (Campbellfield, VIC, Australia). The cutting direction was oriented in such a way to ensure that when the test coupons have assembled the landing of the prepared welding joint

would remain perpendicular to the rolling direction of the plate emulating field conditions. Sandblasting with the ToolTec Sandblast SB-350 (Quebec City, Quebec, Canada) using glass beads, with a diameter up to 305 μm, was performed on the parent plate sections to remove any mill scale and/or surface contaminants.

The V-groove surfaces of the test specimen and 25 mm either side of the weld centreline was polished down with ISO 80 grit (201 μm) emery paper and degreased with acetone. The entire specimen was degaussed prior to welding. The chemical composition of the parent plate and electrodes are given in Table 3 (Bisalloy 80), Table 4 (GMAW-P), and Table 5 (SMAW), respectively.

**Table 3.** Chemical composition of Q&T steel (% Weight) (from Bisalloy Steel Group Limited, Wollongong, NSW, Australia).

| C | P | Cr | Si | S | Al | Ni | Mn | Mo | Cu | Ti | Sn | B | CE$_{(IIW)}$ |
|---|---|---|---|---|---|---|---|---|---|---|---|---|---|
| 0.17 | 0.018 | 0.2 | 0.21 | 0.004 | 0.035 | 0.017 | 1.37 | 0.2 | 0.026 | 0.019 | 0.002 | 0.0017 | 0.4812 |

**Table 4.** Chemical Composition of the AWS A5.28, ER110S-G (% Weight) electrode batch (from Lincoln Electric, Cleveland, OH, USA).

| C | Si | Mn | Ni | Mo | Cr | V |
|---|---|---|---|---|---|---|
| 0.09 | 0.5 | 1.6 | 1.4 | 0.25 | 0.3 | 0.09 |

**Table 5.** Chemical composition of the AWS 5.5 E11018M-H4 (% Weight) electrode batch (from Lincoln Electric, Cleveland, OH, USA).

| C | Mn | P | Si | Mo | S | Ni | Cr | V |
|---|---|---|---|---|---|---|---|---|
| 0.06 | 1.5 | 0.015 | 0.4 | 0.4 | 0.01 | 2.2 | <0.15 | 0.08 |

## 3. Results and Discussion

### 3.1. Welding Mechanics

3.1.1. Speed Trial

Speeds trials were performed at three target heat inputs of 0.5, 1.5, and 2.0 kJ/mm to simulate the widest practicable heat input ranges that would be encountered during the fabrication of safety-critical structures. A maximum difference of ±0.2 kJ/mm was noted between the targeted and actual measured heat input over the range of heat input tests. Additionally, the maximum variation of 4.26% were noted between two welding processes (GMAW-P and SMAW) at the 0.5 kJ/mm target.

For the GMAW-P process, an increase in the heat input, from 0.5 kJ/mm to 1.5 kJ/mm, resulted in a 50% reduction in the number of weld runs deposited, and a further increase in heat input of 0.5 kJ/mm (2.0 kJ/mm) resulted in a further 17% decrease in the number of runs, Figure 6. At a heat input of 2.5 kJ/mm, successful deposition was not possible as a result of consistent burn through.

A similar trend was observed for the SMAW process. However, significant changes in the number of runs were noted as a result of heat input variations. An increase in heat input from 0.5 kJ/mm to 1.5 kJ/mm led to 74% reduction in the number of runs deposited. Additionally, a further increase in the heat input to 2.0 kJ/mm resulted in a 25% reduction in the number of runs. When compared to the GMAW-P, the relationship between number of runs deposited and heat input was nonlinear.

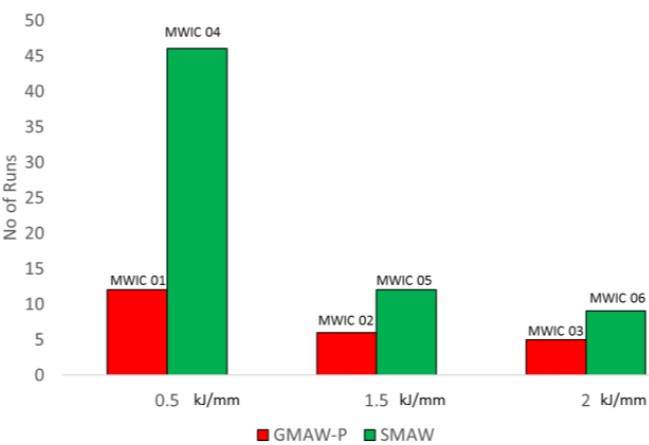

**Figure 6.** Comparative analysis of the recorded number of runs at each target heat input for both SMAW and GMAW-P welding processes. For both welding processes, the number of runs decreased proportionally.

### 3.1.2. Arc on Time (AOT)

The reduction in the number of runs with increasing heat input has indicated itself on the arc on time (AOT). The effect of the number of runs deposited between each process on the total AOT is illustrated in Figure 7.

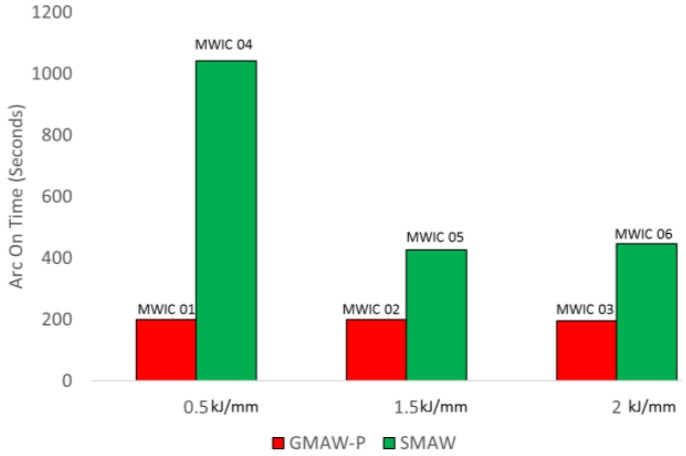

**Figure 7.** Arc on time recorded at each target heat input for both GMAW-P and SMAW welding processes. For the SMAW welding processes, the arc on time decreased dramatically.

For the GMAW-P process, there is a near negligible change in the measured AOT as the heat input is varied from 0.5 to 2.0 kJ/mm. This could be attributed to the proportional increase in the size of fusion zone (melt pool) with increasing heat input.

For the SMAW welding process, however, a change in heat input from 0.5 kJ/mm to 1.5 kJ/mm resulted in a decrease of 59% in the total AOT. A subsequent increase in heat input to 2.0 kJ/mm, however, resulted in a significantly lower reduction in AOT of about ~5%. When considering the geometry of the weld bead deposited, there was a significant difference in the bead height and width between 0.5 kJ/mm and 1.5 kJ/mm. For weld passes deposited at the low heat input of 0.5 kJ/mm, the weld bead width was narrow, and the crown of the bead was pronounced. Subsequent passes had to be deposited in a sequence complementing the existing profile without risking lack of inter-run fusion while maintaining high travel speeds to meet the target heat input. In addition to the lower volume of weld metal deposited at low heat inputs, the geometric characteristics of the

deposited bead can also account for the significantly higher number of runs deposited and higher arc on time (AOT) at the target heat input.

It must also be noted that shielding the higher energy in axial spray transfer inherent with the 92% Ar + 8% $CO_2$ shielding gas employed in the GMAW-P increases puddle (weld pool) fluidity and thus result in a flush face of the crown of the bead, allowing for optimisation of the weld sequence.

### 3.1.3. Total Time (TT)

Unlike AOT which measures the time weld deposition is in progress, the total time (TT) in the context of this body of work, measures the time between the commencement to the completion of the welding activities. This time does not include initial preparation or tacking of the specimens but does include cleaning and changing of electrodes and inter-pass slag removal in the SMAW process, Figure 8.

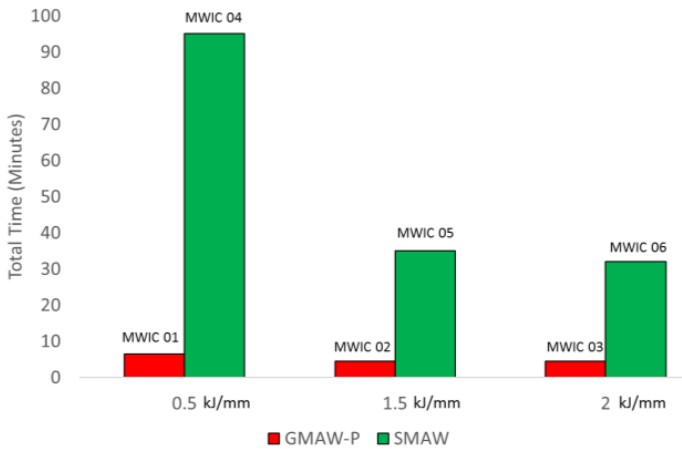

**Figure 8.** Comparison of the recorded total welding time at each target heat input for both the SMAW and GMAW-P welding processes. For the SMAW welding process, the total welding time decreased.

As observed with the AOT, a change in heat input resulted in a marginal change in the total time for the GMAW-P process across the target heat input range. However, when comparing processes, as illustrated in Figure 8, a notable significant difference can be seen in the total time taken to complete fabrication. On average, at 0.5 kJ/mm, there was a 93% difference in the TT for deposition between the SMAW and GMAW-P processes. For the heat input targets of 1.5 and 2.0 kJ/mm, there was a difference of 87% and 85%, respectively.

### 3.1.4. Productivity Gains

From the speed trials, there is sufficient evidence to summarise that the productivity grains, in terms of (a) the reduction in the arc on time as a result of the reduced number of runs, (b) the reduction in the total number of runs required to complete the joints, and (c) the total time saving from employing GMAW-P makes its adoption in the welding of thick sections of Q&T steels favourable. On average, adoption of the GMAW-P lead to a reduction of 63% in the 'Arc-On' time and 88% in total deposition time over the heat input range tested. Noting that all the tests were conducted in the Flat (1G) position. The trials clearly demonstrate that there is a significant improvement in productivity when employing GMAW-P.

### 3.2. Welding Defects

The application of MWIC weldability tests enabled us to study the formation of potential defects that may occur during the welding of Q&T steel, Bisalloy 80. In general, there are two major discontinuities that could occur in welded structures:

(1) Hydrogen crack-(HACC), mainly forms at low temperatures due to the diffusion of hydrogen and reduction in hydrogen solubility due to microstructural transformation.

(2) Cracks attributed to the lack of fusion during welding.

To examine the susceptibly of the weld metal and HAZ to HACC, transverse sections of the welds deposited on the MWIC weldability test were examined under ×400 optimal magnification [38] and presented in Figure 9. The examination did not yield any evidence of hydrogen cracking, suggesting that under high levels of restraint, as simulated by the MWIC test, both processes were equally resistant to hydrogen cracking under the employed welding procedure.

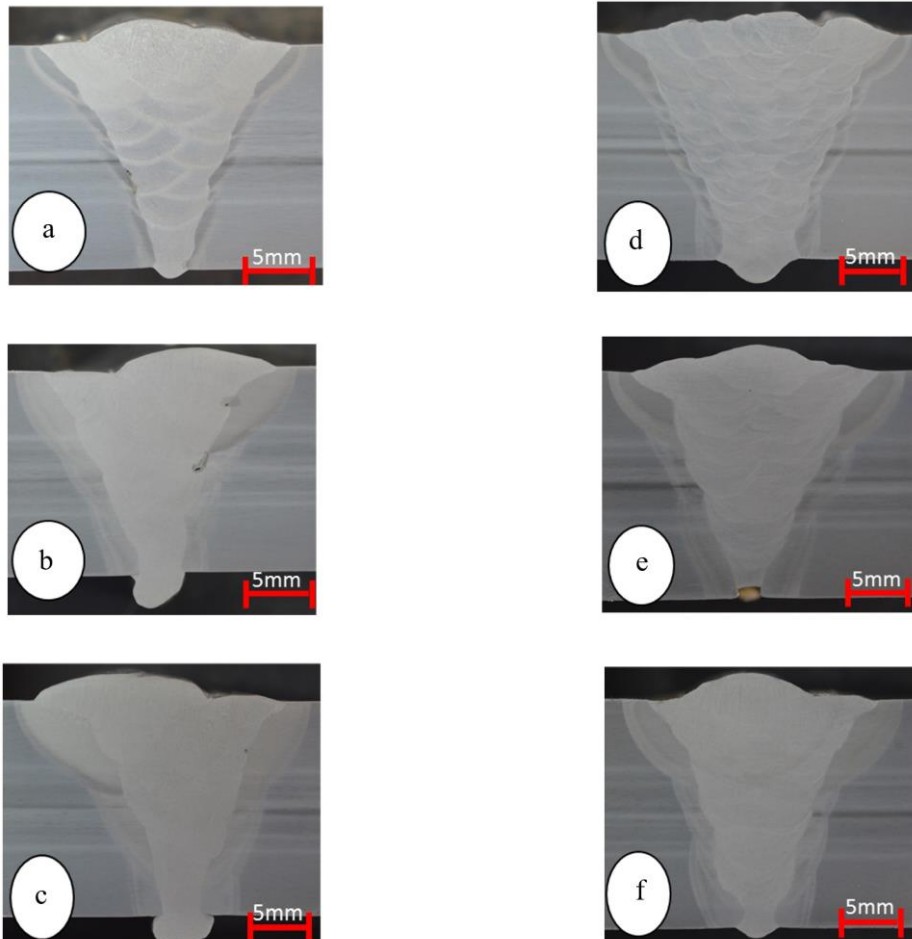

**Figure 9.** Cross-sectional macrograph of the MWIC samples with different heat input; (**a**) MWIC 01 GMAW-P 0.5 kJ/mm (**b**) MWIC 02 GMAW-P 1.5 kJ/mm (**c**)MWIC 03 GMAW-P 2.0 kJ/mm (**d**) MWIC 04 SMAW- 0.5 kJ/mm (**e**) MWIC 05 SMAW 1.5 kJ/mm (**f**) MWIC 06 SMAW-P 2.0 kJ/mm.

However, there were a number of fusion-dependent defects that were identified in the welds deposited using the GMAW-P process. As highlighted in Figures 10 and 11, there was evidence of lack of sidewall fusion for welds deposited at 0.5 kJ/mm and 1.5 kJ/mm.

The lack of fusion may be due to the metal transfer mode where the globular mode of metal transfer is replaced with metal spray transfer. Basically, due to advancements in electronics, the process anticipates and controls the short circuit and reduces the welding current to create a consistent metal transfer. In fact, this reduction in the applied current and instant heat input may be the cause of this lack of fusion as the weld pool is coming into contact with much cooler parent metal. This is indicated in the extent of HAZ when its thickness in Figure 9a is compared with HAZ thickness in Figure 9d. The much cooler weld metal adjacent to PM could be the root cause of the lack of fusion. The formation of fusion-related defects is evident mainly near the sidewall as depicted in Figures 10 and 11.

There is, however, odd inter-run weld defects indicating that the lack of fusion could occur within the body of the weld joint between the weld passes as shown in Figure 10a, ROI 4.

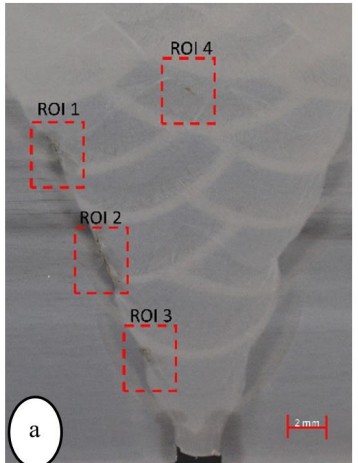
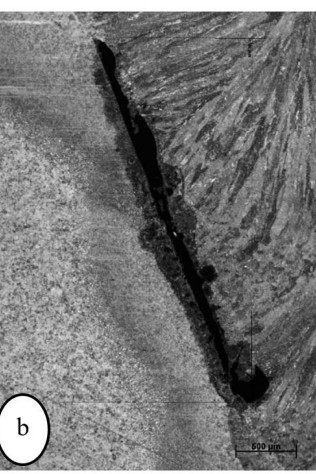

**Figure 10.** (**a**) Lack of sidewall fusion (ROI 1-3) and lack of inter-pass fusion (ROI 4), noted in MWIC 01, GMAW-P 0.5 kJ/mm and (**b**) lack of sidewall fusion (ROI 2 with higher magnification).

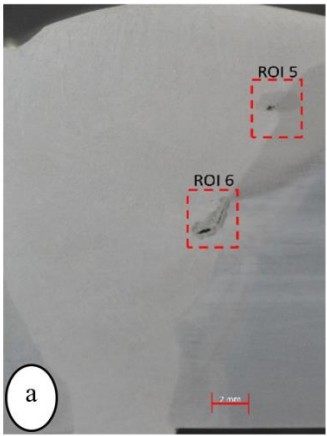
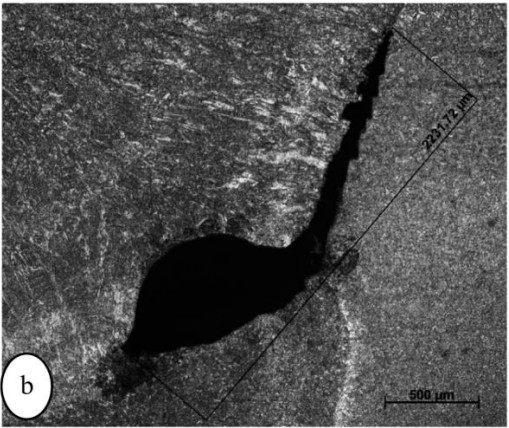

**Figure 11.** (**a**) Lack of inter-run fusion and a slag inclusion (ROI 5) and lack of side wall fusion (ROI 6), noted in MWIC 02, GMAW-P 1.5 kJ/mm and (**b**) lack of side wall fusion with higher magnification.

*3.3. Microstructure*

Detailed microstructural analysis was carried out for different zones of PM, HAZ, and WM of the samples welded with GMAW-P and SMAW. The fine-grained microstructure of the as-received Q&T AS/NZS 3597 Grade 700 steel (Bisalloy 80) is mainly granular bainite, as shown by the optical micrograph in Figure 12. Figure 13 shows the optical micrographs taken from the coarse-grained heat-affected zone, CGHAZ, and fine-grained heat affected zone, FGHAZ, regions in the mid-plate thickness of the both GMAW-P and SMAW-welded samples. The FGHAZ region for both welds appeared to have developed a mainly ferritic (white phase)-pearlitic (dark region) structure with some residual bainitic grains of the PM, Figure 13b,d,f,h. However, the morphology of the ferrite phase is slightly different at low and high heat inputs; at 0.5 kJ/mm, the ferrite phase has resemblance to bainitic laths which is referred to as "bainitic ferrite", while at high heat input of 1.5 kJ/mm, the ferrite tends to be more developed towards a polygonal morphology. The formation of pearlite may suggest the HAZ temperature has increased above the eutectoid temperature during welding enabling the formation of ferrite and pearlite upon cooling within the FGHAZ region. There is an increase in the grain size with increasing heat input from 0.5 kJ/mm to 1.5 kJ/mm as can be seen in Figure 13d,h. The morphology of ferrite resembles that of bainitic lath with clear coarsening. The CGHAZ region showed large bainitic grains at low

heat input, Figure 13c, and mainly contains polygonal ferrite (while phase) and bainitic ferrite (white phase with some fine particles) at a high heat input of 1.5 kJ/mm as shown in Figure 13e. The CGHAZ for the SMAW samples show similar microstructure as GMAW-P at lower heat input of 0.5 kJ/mm, but it is more bainitic and even some martensite patches may be seen with higher heat input SMAW samples. Such a change is expected to show itself up in the hardness measurement as reported later in this report.

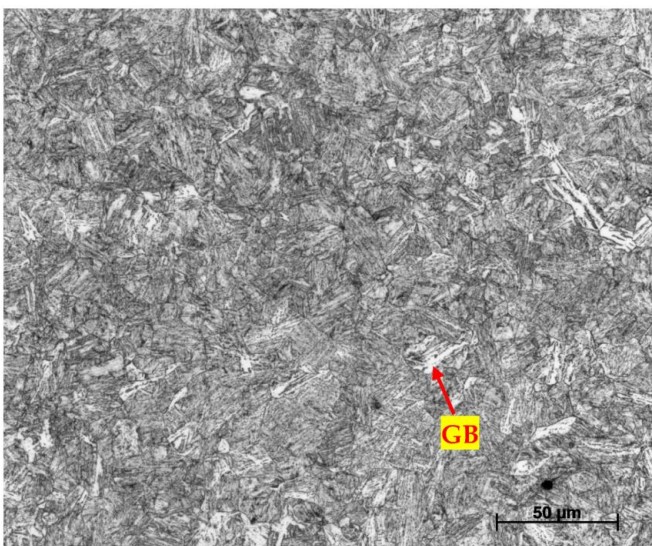

**Figure 12.** Optical micrograph showing the PM of Q&T AS/NZS 3597 Grade 700 steel (Bisalloy 80) with mainly granular bainite (granular bainite: GB).

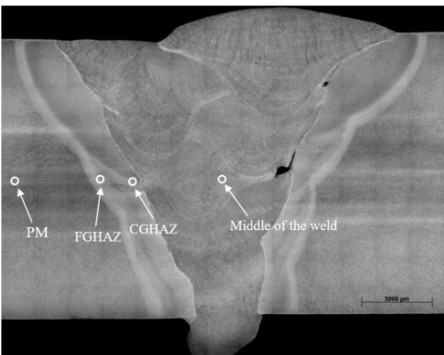

(a) Locations of the optical micrographs taken for the PM, HAZ and WM

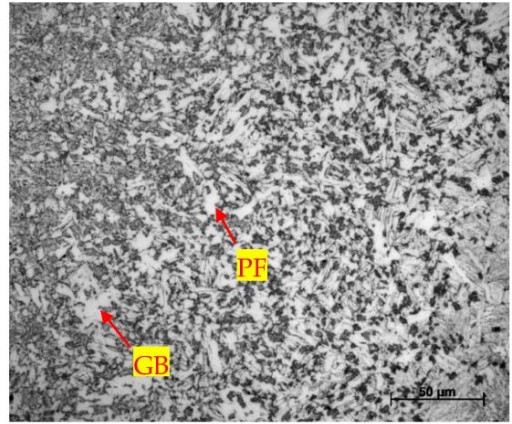
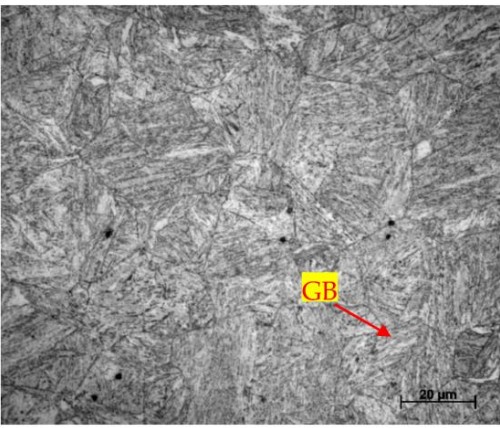

(b) FGHAZ (MWIC 01 GMAW-P 0.5kJ/mm)      (c) CGHAZ (MWIC 01 GMAW-P 0.5kJ/mm)

**Figure 13.** *Cont.*

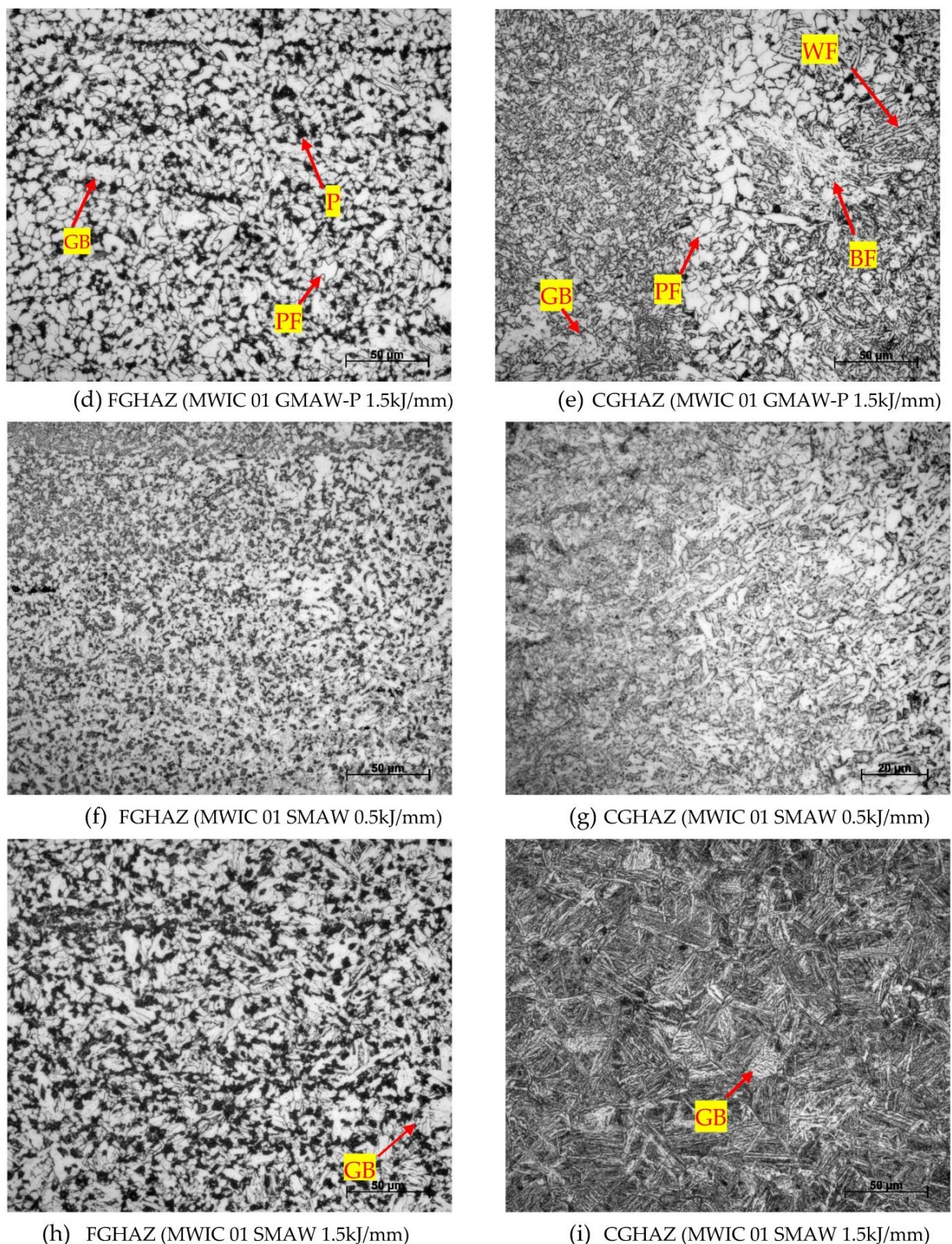

**Figure 13.** Optical macrograph showing the location of micrographs for different regions of the PM, HAZ, and WM (**a**); optical micrographs in the mid-plate thickness showing the HAZ, CGHAZ and WM for the GMAW (**b**–**e**); and SMAW (**f**–**i**) of the MWIC specimens (BF: bainitic ferrite; GB: granular bainite; PF: polygonal ferrite; P: perlite; WF: Widmanstätten ferrite).

The optical micrographs for the weld metal (WM) are shown in Figure 14. The weld metal microstructure for the SMAW and GMAW at low heat input is mainly acicular ferrite with some Widmanstätten ferrite. There is, however, microstructural changes for the specimens with higher heat input with the formation of polygonal ferrite along with the mixed phases of acicular, bainitic, and Widmanstätten ferrites, Figure 14b,d. Looking purely visually at the WM images for both GMAW and SMAW welds, it is also noticeable that the WM of the both processes with higher heat input, Figure 14b,d, has a coarser microstructure, acicular ferrite, and larger grain size, compared to the one with the lower heat input. This is expected as the cooling rate decreases with increasing heat input, since the heat sink is the same for both tests.

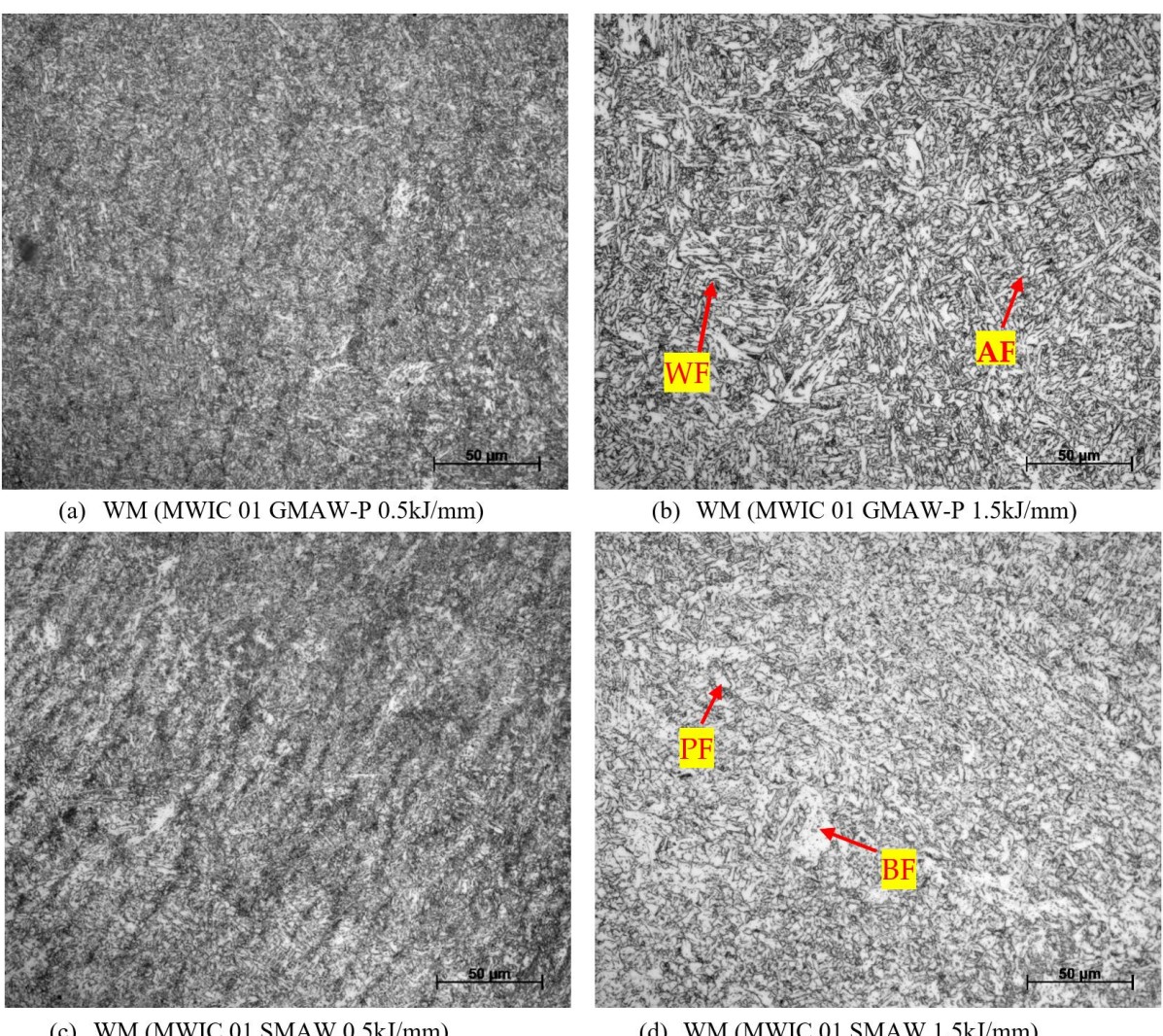

(a) WM (MWIC 01 GMAW-P 0.5kJ/mm)

(b) WM (MWIC 01 GMAW-P 1.5kJ/mm)

(c) WM (MWIC 01 SMAW 0.5kJ/mm)

(d) WM (MWIC 01 SMAW 1.5kJ/mm)

**Figure 14.** Optical micrographs showing the WM for the GMAW-P (**a,b**) and SMAW (**c,d**) of the MWIC specimens (AF: acicular ferrite).

### 3.4. Weld Qualification-Mechanical Testing

To assess the comparability of the welding processes, from an industrial application perspective, two coupons were welded in accordance with AS/NZS 1554.4:2014 (SP) at the target heat input of 1.5 kJ/mm using the parameters listed in Tables 1 and 2. It is critical to note that a preheat of 25 °C was applied and a maximum inter-pass temperature of 150 °C was maintained. A comparable weld sequence was maintained between processes; however, the number of runs differed as expected and discussed earlier.

### 3.4.1. Hardness

A Vickers hardness survey was conducted in accordance with AS2205.6.1:2003 using a square-based diamond indenter. Three hardness traverses were conducted to assess the hardness profile across the transverse section of the ligament, for the subsequent layer of weld metal. In the first traverse, Figure 15a, a peak hardness of 318HV10 was measured in HAZ of the GMAW-P process and 349HV10 in the HAZ for the SMAW process. For the second hardness traverse, Figure 15b, maximum hardness of 305HV10 was measured in HAZ of the GMAW-P process and 279HV10 in the HAZ for the SMAW process. For the third hardness traverse, Figure 15c, a maximum hardness of 305HV10 was measured in HAZ of the GMAW -P process and 279HV10 in the HAZ for the SMAW process. With the exception of the bottom traverse where a mean difference in hardness of approximate 20% was noted, there was a marginal difference (<2.5%) in mean hardness values measured across the weld.

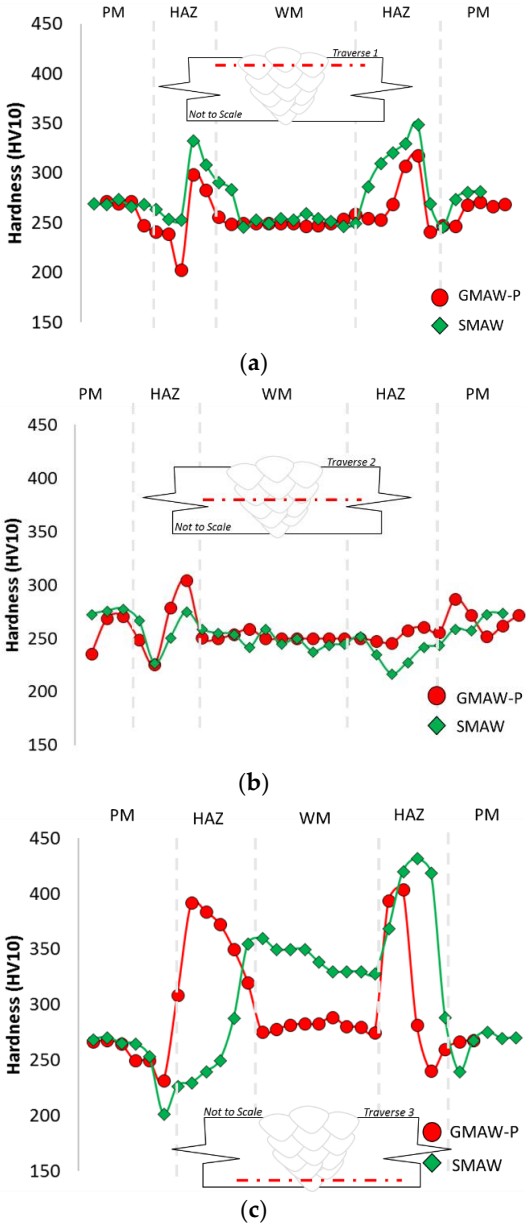

**Figure 15.** Hardness traverse plot in the (**a**) top third of the weld bead; (**b**) mid-plate thickness; and (**c**) bottom of the weld for both the GMAW-P and SMAW processes.

Higher hardness observed in the HAZ of both processes could be attributed to the type and nature of the phases and their degree of refinement or coarsening in contrast to weld metal. The lower hardness values experienced in the weld metal for both SMAW and GMAW-P processes, in comparison with the HAZ, may also be associated with the tempering effects due to multi-pass deposition. This is more evident for the mid-plate thickness layers of the welds (Figure 14b,d), which leads to the formation of tempered bainite, and polygonal ferrite as well as causing coarsening of the acicular ferrite in the weld zone.

### 3.4.2. Impact Toughness Tests

Weld metal impact testing was carried out in accordance with AS 2205.7.1:2003. The impact specimen size was 10 mm × 10 mm, and a notch depth of 2 mm was recorded. A total of 3three impact tests were taken for each weld process, and a strike force of 406 J was applied in each test. The mean impact strength was 98 J and 69 J for the SMAW and GMAW-P processes, respectively, Figure 16. There was also a greater variance in the measured impact strength for welds deposited using the GMAW-P process. High variation in the measured impact strength with lower mean for GMAW-P process, in comparison with SMAW process, could be related with the formation of defects (i.e., lack of inter-run fusion and slag inclusion) in the HAZ and the weld metal adjacent to the fusion zone as illustrated in Figure 11.

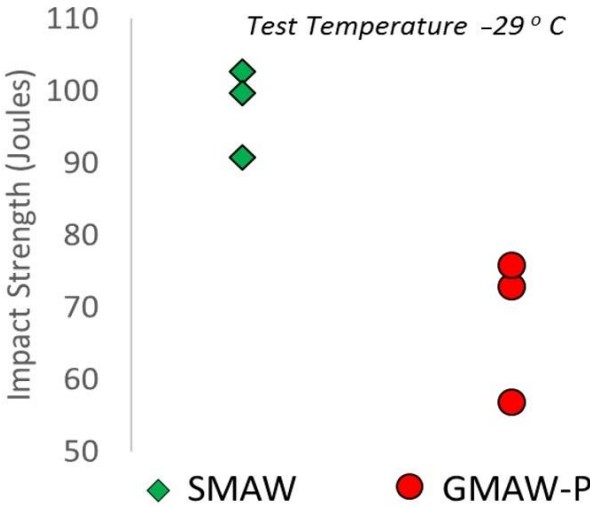

**Figure 16.** Weld metal Charpy V notch impact test results for 10 mm × 10 mm impact size specimens.

### 3.4.3. Transverse Tensile Tests

Transverse tensile testing was carried out in accordance with AS2205.2.1:2003. Two tensile tests were conducted on each coupon. As illustrated in Figure 17, there was less than 3% difference in the measured UTS. Additionally, fracture occurred in the weld metal for samples from both processes. It has to be mentioned that the tensile samples for the GMAW-P were prepared from regions that did not contain any lack of fusion defects. That might be the reason for GMAW-P welds having similar tensile strength as SMAW-welded samples. It is expected that the lack of fusion near the HAZ region will have detrimental effects on tensile properties.

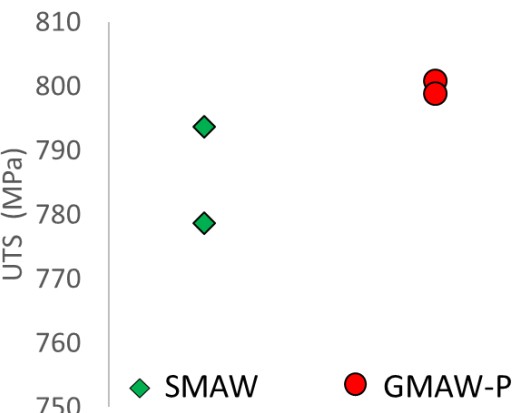

**Figure 17.** Transverse tensile tests results for 30 mm × 19.5 mm coupons. An average ultimate tensile stress 786.5 MPa and 800 MPa were recorded for the SMAW and GMAW-P.

3.4.4. Transverse Bends

Transverse side bend tests were conducted in accordance with AS 2205.3.1:2003. Two side bend tests were conducted in each process, to determine crack formation and if there is any contribution from lack of fusion defects. No apparent detrimental effects were noticed with respect to lack of fusion defects in samples from either process. This could be related with the size of the defects for the MWIC sample with the heat input of 1.5 kJ/mm (the selected samples for the mechanical testing), which have minimal impact on the crack formation, see Figure 11. It may also be related to the orientation of lack of fusion cracks with respect to loading orientation.

The mechanical tests serve as a clear demonstration that a well-designed procedure can yield comparable mechanical properties for welds deposited using both welding processes. The main advantage of employing GMAW-P is embedded, in the context of this body of work, in the productivity gains, in particular, the reduction in arc on time (AOT) and total time (TT), which has a significant impact on the economic feasibility of adopting the GMAW-P process.

It may be argued that the difference in impact strength, which is of high importance in structures fabricated from quenched and tempered steels, are a cause for concern. It is believed that with proper selection of welding parameters in conjunction with appropriate consumable chemistry, the formation of defects due to the lack of fusion could be minimized to allow full advantage of economic gains by using the GMAW-P welding technique.

In the future extension of this work, a range of consumable compositions will be tested in conjunction with several welding parameters to examine how the process can be optimised to achieve the desirable impact strengths.

*3.5. GMAW-P vs. SMAW*

When considering the gains in productivity, the lower weld metal impact toughness and the propensity to lack of fusion-type defects, suggest that further work is needed to optimise weld procedure development before modified pulse gas metal arc welding, GMAW-P, can be used as a successful alternative to SMAW when welding quenched and tempered steels for safety-critical structures. While welding procedure development can be optimised to reduce the number of imperfections and defects, there remain fundamental limitations of the process. With GMAW-P, the ability to consistently deposit homogeneous weldments is limited. There is sufficient anecdotal evidence in the industry that demonstrates the presence of lack of fusion and porosity-type defects in GMAW and GMAW-P process weldments. Welding codes (such as AS/NZS 3992 and ASME BPVC IX) acknowledge these limitations by specifying transverse, face, and root bend testing of GMAW weldments in addition to radiography. The results generated in this body of work support

these observations. Production conditions will be more onerous than the test conditions used for this work.

Production conditions and the variable skills of welders engaged should be considered in the selection of welding process for safety-critical structures. It should be noted that the active flux elements in SMAW electrode coatings, in particular, the scavenging elements, help produce welds with reduced porosity.

Although SMAW is a preferred process in terms of microstructural homogeneity due to retarding the cooling rate and improving the solidification structure, but adoption of this process is associated with extra cost and lower production rate.

## 4. Conclusions

The techno-economic feasibility of GMAW-P for the welding of thick sections of Q&T steels (AS/NZS 3597 Grade 700) was assessed through comparing deposition rates, microstructural and mechanical properties, and susceptibility to HACC for a range of heat inputs to the conventional SMAW process. The key finding of this experimental study were

(1)   Adoption of GMAW-P resulted in an average reduction of 63% and 88% in the 'Arc-On' time and the total normalised fabrication time, respectively, for the heat input range (0.5–2.0 kJ/mm) tested condition.

(2)   Weldability testing on the MWIC test demonstrated that for the selected filler materials (E11018M-H4 and ER 110S-G), under high restraint (Rf = 25 mm) and within the target heat input range, the GMAW-P did not show an increased susceptivity to cracking. However, at low heat inputs (0.5 kJ/mm), welds deposited using GMAW-P was prone to lack of inter-run fusion and lack of sidewall fusion.

(3)   The weld metal and HAZ regions for both GMAW-P and SMAW have comparable microstructure with similar mechanical properties with respect to tensile strength and hardness while the SMAW welds show superior toughness. The main drawback with GMAW-P is the formation defects, mainly at regions near fusion zone–HAZ interface, due to the lack of fusion between the weld metal and parent metal.

More work, consumable chemistry and welding parameters, is required to demonstrate the productivity improvement while maintaining dependability in the deposition of homogeneous weldments with gas-shielded wire processes.

**Author Contributions:** Conceptualization, H.A., N.C., R.G.; methodology, H.A.; validation, H.A. and N.C.; formal analysis, H.A.; experimental investigation, H.A., N.C., R.K. and A.R.; resources, R.G. and N.C.; data curation, H.A. and R.K.; writing—original draft preparation, H.A.; writing—review and editing, R.G.; visualization, H.A. and R.K.; supervision, R.G.; project administration, H.A., R.G. and N.C.; funding acquisition, R.G. and N.C. All authors have read and agreed to the published version of the manuscript.

**Funding:** This research received no external funding.

**Data Availability Statement:** Not applicable.

**Acknowledgments:** The welding operation was performed at the Australian Welding Solutions. The authors would like to thank Damien Lynch, AWS, and Pascal Symons, School of mechanical Engineering of the University of Adelaide for their expert welding of test coupons.

**Conflicts of Interest:** The authors declare no conflict of interest.

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
