# Peer review of "Quenched and Tempered Steels Welded Structures: Modified Gas Metal Arc Welding-Pulse vs. Shielded Metal Arc Welding"

_metals, doi:10.3390/met13050887_

Round 1

Reviewer 1 Report

The data and observations presented in this manuscript  provide the basis for an acceptable paper but the authors must respond to the following questions and comments.

1. Figure 1 is difficult to understand. It is unclear in this perspective drawing how this arrangement results in constraint on the weld sample. Perhaps a short paragraph giving a brief description would be appropriate. Also, it is difficult to read the labels because the font is too small.

2. In Table 1, what are the voltages and currents associated with the pulse mode of operation in the GMAW-P? In Table 2, what is the electrode coating for the SMAW procedure? In Table 3, what is the thermomechanical history of the base plate? At least, what are the final heat treatment parameters producing the Q&T condition and the microstructure of Figure 14?

3. A schematic or macrophotograph labeled to show the locations of the microstructures illustrated in Figures 15 and 16 is needed. 

4. Also, the microstructures in Figures 15 and 16 should be correlated with the hardness traverses in Figures 17 and 18.

5. Finally, was any attempt made to evaluate the dilution effects, for example of Ni, from weld metal into base metal? If so, did the welding processes have any effect on dilution?

Reviewer 2 Report

1. The abstracts should be combined into one paragraph.

2. The formats of the manuscript are not correct for a research paper, for example, table title should be put above the table, out line of figure captions should be deleted, sub figure should be numbered as (a), (b)...in Fig. 11, 12 and 13.

3. Fig. 6  and Fig. 7 should be deleted.

4. Please indicate the typical phases such as ferrite, bainitic and Widmanstätten ferrites in Fig. 14 to 16 using arrows.

5. Please combine Fig. 17 to Fig. 19 into one Figure.

6. Please give the reason why the impact strength of GMAW-P joint is lower than that of SMAW joint.

Reviewer 3 Report

There were some questions as follow.

1.     What was the research contents on Hydrogen Assisted Cold Cracking (HACC)? And what was the relationship between the parameters that were measured and HACC?

2.     In Figure 14, what was the difference between welded zone and base metal by optical micrograph?

3.     Transverse Tensile Tests were finished, and how about tensile fracture morphology and their difference?

Round 2

Reviewer 1 Report

The authors response to my comments and critical remarks is very good. The paper is now acceptable for publication.

Reviewer 2 Report

The manuscript has been revised according to my comments. It can be accepted now.

Reviewer 3 Report

It can be accepted.